# Selection Response in a Divergent Selection Experiment for Birth Weight Variability in Mice Compared with a Control Line

**DOI:** 10.3390/ani10060920

**Published:** 2020-05-26

**Authors:** Nora Formoso-Rafferty, Katherine Natalia Chavez, Candela Ojeda, Isabel Cervantes, Juan Pablo Gutiérrez

**Affiliations:** 1Departamento de Producción Agraria, E.T.S. Ingeniería Agronómica, Alimentaria y de Biosistemas, Universidad Politécnica de Madrid, C/Senda del Rey 18, 28040 Madrid, Spain; nora.formosorafferty@upm.es; 2Departamento de Producción Animal, Facultad de Veterinaria, Universidad Complutense de Madrid, Avda. Puerta de Hierro s/n, 28040 Madrid, Spain; kth.chavez@hotmail.com (K.N.C.); candelao@ucm.es (C.O.); gutgar@vet.ucm.es (J.P.G.)

**Keywords:** birth weight, divergent selection response, mouse, robustness, variability

## Abstract

**Simple Summary:**

The divergent selection for birth weight variability was previously proven by the difference between the two selected mice lines, but the comparison with a control line made it possible to measure independently the differential response in each of the low and high birth weight variability lines. In conclusion, the genetic response was much higher for the low variability line than for the high variability line. This was extremely satisfactory given that homogeneity provides advantages in terms of animal welfare and robustness.

**Abstract:**

Birth weight (BW) in animal production is an economically important trait in prolific species. The laboratory mouse (*Mus musculus*) is used as an experimental animal because it is considered a suitable model for prolific species such as rabbits and pigs. Two mouse lines were divergently selected for birth weight variability with a third line of non-selected control population of the same origin as the animals starting the experiment. The objective of this study was, therefore, to compare and evaluate the differential response of each line. The animals were from the 17th generation of both low and high BW variability lines of the divergent selection experiment, including in addition animals from the control line. The dataset contained 389 records from 48 litters of the high line, 734 records from 73 litters of the low line, and 574 records from 71 litters of the control line. The studied traits were as follows: the BW, the BW variance, the BW standard deviation, the BW coefficient of variation within-litter, the weaning weight (WW), the litter size at birth and at weaning, the weight gain, and the preweaning survival. The model included the line effect jointly with the parturition number and its interaction, the linear and quadratic LS as covariates except for the LS trait itself when analyzing litter traits, as well as the pup sex when analyzing individual traits. The low line had a lower BW and WW, but a higher litter size, and greater robustness owing to a higher survival at weaning. As a model of livestock animals, the findings from this experiment led to a proposal of selection for pig production that would combine an increase in litter size with higher survival and welfare. Compared with the control line, a much higher response was observed in the low variability line than in the high line, making it extremely satisfactory given that homogeneity provides advantages in terms of animal welfare and robustness.

## 1. Introduction

The genetic control of the environmental variability in the search for a higher homogeneity is becoming important in current animal breeding programs [1,2,3]. Selection for homogeneity would result in animals that are more robust and better prepared to face any environmental challenges [4]. Moreover, enhancing robustness will improve the functional traits while maintaining as much of the high production potential as possible [5]. Recently, some authors have proved that selecting to reduce the environmental variability of a particular trait was possible [6], thus leading to more robust animals [7]. In the context of the animal production industry, homogeneity has also been associated with a reduction of handling and production costs, resulting in homogeneous batches of animals, fewer cases of mortality in prolific species, and higher financial results [8]. Other authors also reported that animals in heterogeneous litters were more prone to diseases affecting siblings in the same litter [9]. Decreasing the environmental variance also increases heritability, which is particularly interesting because, in the selection response, this can lead to an increase in traits with low heritability, such as litter size in prolific species like rabbits [10] and mice [11].

Formoso-Rafferty, et al. [12] developed a divergent selection experiment for birth weight variability in mice and they concluded that it was possible to modify the genetic control of the birth weight environmental variability. They also showed that this selection criterion had direct effects in other interesting traits in livestock such as litter size, weaning weight, and survival [6]. As a result of the divergent experiment, two divergent lines were created, with the low line presenting benefits in production [12], animal welfare [6], heritability [11], and robustness traits [13,14].

Therefore, selecting to reduce the variability is also a benefit. Divergent selected lines performed oppositely regarding several traits after selection and a comparison with a control line could clarify whether or not these selection responses were symmetric in both lines. The objective of the present study was to compare the genetic response of two divergent selected lines for birth weight environmental variability with a non-selected control line. Part of this objective was also to analyze their differences for birth weight variability, birth and weaning weights, litter size, and preweaning survival.

## 2. Materials and Methods

### 2.1. Data

The three experimental mice lines included the following: two lines divergently selected for birth weight (BW) environmental variability and a non-selected population used as control population. Data proceeded from the offsprings resulting from the mates between 43 males and 43 females per line of the 17th generation of a running successful divergent selection experiment conducted to modify the environmental variability of BW [12]. Under this selection experiment, the animals were mated following a design determined by simulated annealing that optimized the mean genetic breeding values of the progeny without exceeding the coancestry level determined by a standard solution [15]. Moreover, a restriction of not sharing grandparents was imposed on the animals to be mated. Complete details of the selection process were described in [12]. These lines will be referred to as “low” and “high” for low and high variability lines, respectively, across the rest of the present paper.

Simultaneously, a non-selected population (control line) with the same origin as the selected lines evolved under mating carried out randomly without sharing common grandparents to prevent inbreeding. In the control population, 29 males were mated with 2 females, and 3 males with 1 female, each resulting in 61 females having the chance of giving birth to litters. Eventually, half of the females lodged in pairs with a male of the control line (one per male) were individually lodged on day 20 after mating for handling purposes, keeping the other females lodged with the male. All the females were checked daily during the parturition period, and all the newborns from the females giving litters were individually identified and weighed at birth (BW), except stillborns, which were taken into account to register the litter size (LS) of the female at the parturition. The survival at birth (SB) was also recorded.

At 21 days old, the pups from all the three lines were weaned and sexed, and the litter size (LS) in which they were born was registered. Phenotypic variance (V), standard deviation (SD) and coefficient of variation (CV) of BW within litter, weight at weaning (WW), preweaning average weight gain (WG), litter size at weaning (LSw), and survival at weaning (SW) were also registered for each pup from all the three lines and afterwards analyzed. The dataset contained 389 records from 48 litters from 37 females of the high line, 734 records from 73 litters from 42 females of the low line, and 574 records from 71 litters from 53 females of the control line. After editing, the data set contained a total of 1697 BW records and 1602 WW records from 192 litters of 132 females. The mean and standard deviation of all the traits are included in Table 1.

The conditions of housing and management of the animals conformed to Spanish legislation RD 53/2013, which includes the basic rules for the protection of animals during experiments and other scientific purposes (BOE, 2013). This experiment was approved by the Ethical Committee of Complutense University of Madrid and authorized by the Regional Government of Madrid (PROEX 308/14 and PROEX 224/2018). The animals were housed in the experimental facility at the Department of Animal Production of the Veterinary Faculty of the Complutense University.

### 2.2. Statistical Model

The BW variability (V, SD, and CV), BW, WW, WG, LS, LSw, SB, and SW were analyzed using a general linear model procedure with the general following equation:(1)y=μ+line+PN+line×PN+sex+LS+LS×LS+e
where y is any of the mentioned traits, μ is the mean, line (with three levels: high, low, or control line), PN is the parturition number (with two levels: first or second parturition), sex (male, female, or unknown; not fitted for BW variability traits, nor for LS and LSw), LS is the litter size at birth (not fitted for LS trait nor for LSw), and e is the residual. Instead of fitting a mixed model with a maternal random effect, the line effect is expected to absorb the genetic differences in this scenario in which the number of records is particularly limited for the control line. Even though the influence of the genetic drift was present, given the random genetic drift direction, it is reasonable to think that, after 17 generations, its cumulated effect would have been roughly similar for all the lines.

The Statistical Analysis System software [16] was used to perform the statistical analysis of the data. The means were compared by the Tukey’s test under different significance levels (*p* < 0.05, *p* < 0.01, and *p* < 0.001). The least square means were computed to show the observed differences between levels of relevant effects.

### 2.3. Selection Intensities and Effective Population Size

Selection intensities and effective population sizes were determined independently for each of the lines to check for possible causes in case the selection response was different in different lines. The final selection intensity was calculated as follows:(2)i=x¯s−x¯cσc
where $\sigma_{c}$ is the standard deviation of the predicted breeding values of the candidates for selection, ${\bar{x}}_{c}$ is the mean predicted breeding values of the candidate animals to be selected, and ${\bar{x}}_{s}$ is the mean of the implemented optimal genetic solution that averages the predicted breeding values of the selected individuals weighted by their offspring contribution.

The effective population sizes were computed from the individual increase in inbreeding [17] and from the increase in pairwise coancestry [18] using the Endog v4.8 software [19].

## 3. Results

Table 2 shows the selection intensities, equivalent proportion selected, and effective population size for each generation and line. The selection intensity was lower in later generations than at the beginning, and this loss in the selection intensity was similar in both lines. The effective population size was lower in later generations owing to the selection process being based in coancestry higher than that based in inbreeding; both selected lines had very similar values.

Table 3 includes the results from the analysis of variance for the effects included in the statistical model: the line, the parturition number, the line × parturition number interaction, the sex, and the linear and quadratic litter size covariates when fitted or not, on V, SD, CV, BW, WW, WG, LS, LSw, SB, and SW. For all the traits measuring BW variability (V, SD, and CV), only the line effect was strongly significant. Other effects fitted in the model did not affect the BW variability.

Figure 1 shows the mean BW variability within litter for the high line, the low line, and the control line. No differences were found between the high and the control lines, but there was a significant difference between these two lines and the low line. The control line was located between the other two lines, but much closer to the high line. The mean BW variance (±standard deviation) for the control was 0.031 g (±0.003), 0.038 g (±0.004) for the high line, and 0.011 g (±0.003) for the low line. The mean BW standard deviation for the control line was 0.156 g (±0.009), 0.180 g (±0.011) for the high line, and 0.097 g (±0.007) for the low line. Finally, the mean BW coefficient of variation for the control line was 0.091 (±0.005), 0.102 (±0.006) for the high line, and 0.067 (±0.004) for the low line. The control line had an intermediate BW variability placed between the high and low line.

Regarding BW, Table 3 shows that it was affected by the line, the parturition number, the line*parturition number interaction, the sex, and the litter size effects. The mean BW according to the parturition number in every line is shown in Table 4. The high and control line presented bigger BW than the low line in both parturitions. Only in the control line did the BW show differences between parturitions, being bigger in the second parturition (1.59 vs. 1.72 g). As for the parameters measuring variability, the means of the control line were much closer to the high line than the low line. Figure 2 shows the mean BW and LS at birth in the three lines. Regarding the LS mean, there were no significant differences between the control line (8.04) and the high line (8.45). In contrast, the low line had a significantly higher LS mean with 10.04 pups.

There were significant differences in WW for the line, PN, and LS quadratic covariate effects (Table 3 and Table 4). The mean WW for each parturition number of every line is included in Table 4, with a slightly higher mean in the first parturition in the three lines.

Figure 3 shows the mean for WW and LSw in the three lines. There were no differences in the mean for WW between the control line (11.38 g) and the high line (11.37 g); this was the same situation as for BW. In contrast, the low line presented a mean for WW significantly lower than the other two lines (9.57 g). Regarding LSw (Figure 3), there were no differences between the control line (6.90 pups) and the high line (7.26 pups). In contrast, the low line had a mean for LSw significantly higher (9.77 pups) than the other two lines.

Although it was expected that the control line would also present an intermediate value for weaning weight, no significant differences were found at weaning between the control and the high line. The low line presented lower WW than the control line and the high line, but larger LSw. Regarding WG, there were differences for the line, parturition number, line × parturition number interaction, and LS quadratic covariate effects (Table 3 and Table 4). The mean WG was higher in the first parturition in the three lines (Table 4). The control line presented the highest mean for WG (10.44 g). The low line performed very differently from both the control and high lines.

The line and LS had significant effects on survival at birth and at weaning (Table 3). According to the linear and quadratic LS covariate estimates, the optimal LS was around 10 from the survival point of view at both birth and weaning (result not shown). Table 5 shows that the low line had a higher survival in both than the high line, while the control line placed between them was not different from any of the divergent lines. The differences in survival between lines led to an LS difference between the low and high lines of 1.59 pups at birth and 2.87 pups at weaning.

## 4. Discussion

The success of the divergent selection experiment for environmental variability of birth weight had been proven for the selection objective [12] and for other correlated traits [6] by comparing each line with each other. However, it was not proven if selection was balanced between the lines, or one of the lines perhaps did not response or the lines had responded with a different magnitude. This is a significant issue, as selection for reducing the variability has been shown to have great benefits, unlike the high line variability [12,13,14]. This was possible to check through the control line. For the analyzed traits addressing BW variability, the control line had intermediate values that fell between the high and low lines, thus suggesting that it was a selection response in both lines. Nevertheless, there were no statistical differences between the high and the control lines, showing a non-symmetric genetic selection response between lines. This asymmetric response was unexpected, as the selection process was designed and carried out identically in both lines. This idea was reinforced by the information gathered during the experiment. For example, the mean predicted breeding values for the animals born in the last generation were 0.61583 for the high line and −0.55037 for the low line, with 0.00454 for those in the generation originating the lines. These values look similar in absolute values, but higher in the high line. It could be argued that, as the high variability line had a poor reproductive performance [6,13], the intensity of selection could have been different between the lines. Table 2, however, shows that neither the intensity of selection nor the effective population size seem to have been affected, as they are very similar in both lines.

Regarding the positive correlated genetic response, the pups from the high line, the heterogeneous one, were higher for birth weight [12]. Tatliyer, et al. [20] showed that modifying the mean of a trait by selection could cause a change in the variability. Further, modifying the variability of a trait might imply a modification of the mean of this trait in the same direction because of the scale effect. Correlated responses in variability were noted when selecting to increase the mean [21], and changes in the mean trait were observed when selecting for variability [12]. Thus, the higher BW of the high line, together with the higher BW variability, might compromise the biological limits of the uterine capacity to keep large fetuses, which might partially explain a limited selection response in the high line. The physical limits of the uterine capacity would act as a natural protection against an excess of variability, thus producing the non-symmetric response [22,23]. There would be more internal competition for space between fetuses in the heterogeneous line than in the homogeneous line [11]. The parturition number was also significant in BW, WW, and WG. It was previously demonstrated that the parturition number directly influenced the BW, in pigs [24,25] and in goats [26]. Krishnan and Daniel [27] observed that, in pigs, some uterine mechanism limits the number of embryos. This limit may be through a process of selection that benefits the development of the more viable embryos. Some of the concerns of limitations in uterine capacity are the death of fetus before farrowing and small birth weights, which results in reduced postnatal survival and growth [28].

The litter size of the control line did not perform as expected, with an intermediate value between the low and the high lines, but this could have been the effect of a slight difference in the age of the animals, which was around three months older in the control line. Argente, et al. [29] carried out a divergent selection experiment for uterine capacity in rabbits, and observed differences between lines in 1.30 kits at birth in favor of the high line, which agrees with the findings here in mice, being associated to an increase in the number of implanted embryos more probably than to an increase of the fetal survival. In the divergent selection experiment carried out, based on this study, it has been shown that BW variability can be modified by selection, assuming the trait to be maternal. There was also a correlated response with the litter size, with 2.90 more pups in the sixth generation in the low variability line [12]. One of the reasons would be that litter size depends on embryo competence after implantation [30]. Moreover, Argente, et al. [31] performed a selection study for environmental variability of litter size in rabbits for seven generations, and found that the low variability line had a higher litter size. Similarly, Blasco, et al. [7] found better results for litter size in the low variability line than in the high variability line. In addition, the litter size had a positive genetic correlation with the ovulation rate (0.34) and with embryo survival (0.59), and had a high correlation with the number of implanted embryos estimated as the number of implantation sites (0.71) and the survival of the fetus [29]. The selection performed to increase litter size led to the development of hyper prolific sows that ovulate more oocytes as their uterus is capable of housing a large number of fetuses. In the selection experimented base of this work, using ultrasound scans performed at day 14 of gestation showed that embryo mortality was significantly lower in the low variability line than in the high variability line (1.39 vs. 2.87 fetus), but the differences in ovulation rate have not yet been compared between these lines [13].

A smaller BW for the low line can be associated with a higher litter size, but having high prolificacy litters, leading to a higher number of piglets with low BW [25]. Quiniou, et al. [32] showed that the average body weight of piglets could decrease and that the proportion of piglets with low BW can increase with an increase in LS. It was shown that a higher litter size was related to lower birth weight, but it is important to note that, unlike the usual scenario in pig breeding, the litter size was not directly selected, but as a correlated response when selecting for homogeneity [12].

The differences between lines in the weight at birth were also present at weaning, with low WW of animals in the low line, firstly because of the lower BW, but probably also owing to the higher litter size of the low line; the amount of milk the mother produces has to be distributed among more pups. Sows with larger litters produce more milk, but the extra milk is not proportional to the number of extra piglets [33].

The observed relationship among the different traits referred to in the present study could be interesting for pig breeding programs. As consequence of increasing LS, piglets with BW below the litter average could die more than their siblings would, but if they did survive, their weight gain would eventually be similar to the rest of the litter [24]. However, in the mice selection experiment, with a lower LS at weaning and higher WW of the high variability line, there was more competition among pups in the lactation phase; the larger pups tended to harm the smallest pups, only because the pups’ BW was more heterogeneous. In this case, of course, there was not a compensatory growth of the low line animals to end of similar weight to those of the high line. This would not happen in the homogeneous line, with smaller WW, but the total weight weaned was not lower than in the high line [6]. Thus, an interesting finding of the mice selection experiment was that selecting for homogeneity increased LS. However, this increase, unlike when selecting pigs for LS, was accompanied by a higher survival rate at weaning, in spite of the pups being significantly smaller. The pups from the low variability line presented lower weights at birth and at weaning, but with better results for survival and in total litter weight at birth and at weaning. Therefore, the low variability line was shown to be more robust, performed better in survival at birth and at weaning, and contributed more to animal welfare [6]. In addition, the survival at birth and at weaning was higher in homogeneous litters. In the case of rabbits, it was reported that genetic lines with low variability were more robust, with a higher litter size at weaning and better survival of kits than lines with high variability [34].

The relationship between robustness and homogeneity has also been reported in other species; heterogeneous litters were reported to show higher mortality than homogeneous litters in rabbits [8] and pigs [35].

Traditionally, BW was an early measurable trait that attracted great interest because it had a positive genetic correlation with weights later in life [36]. Therefore, selecting for BW may be indirectly used to improve the survival rate of piglets [37]. Additionally, selecting for the capacity of sows to give birth to homogeneous litters may be beneficial for piglet growth and litter homogeneity at weaning, but probably an increased competition in larger litters. Homogeneous growth of piglets within litters could be established in two very different ways. The piglets can either be nurtured equally or the piglets with the most deviant weight, which are most often the smallest piglets, may die [38]. An increase in litter size was shown to be genetically connected with a decrease in the mean piglet birth weight and an increase in the within-litter variability of birth weight [39]. The type of birth significantly affected BW and generally BW decreased with an increase in litter size. Mortality rates were increased for litters with larger variation, hence a reduction within litter variation of piglet weight at birth reduced mortality [40]. The interesting finding of the mice selection experiment was that selecting for homogeneity increased LS, but this increase, unlike when selecting pig populations for LS, was accompanied by a higher survival rate at weaning, despite the pups being significantly smaller. To summarize, we propose that there should be a change in piglet breeding from increasing LS to increasing homogeneity of BW. The litter size would still increase, though not as much as directly selecting for LS; however, this will still positively affect the number of weaning animals. Taking into account the asymmetric selection response observed in favor the low variability line, the advantages of increasing BW homogeneity would, anyway, take effect soon.

## 5. Conclusions

We demonstrated that the selection for BW variability was successful and non-symmetric, mainly observed by the differences in variability parameters between the lines and the control line, with a remarkable higher response in the low line. The selection process led to a higher litter size at birth and at weaning of the low variability line than in the high variability line. These differences were attributable more to the success of the selection process in the low line than in the high line. From a practical perspective, this looks promising because modifying the variability targets the reduction of the variability and not its increase. Despite having lower birth weights and weaning weights, the low variability line is more robust, with greater litter sizes at weaning. This can be attributed to the larger litter size at birth and to a higher weaning survival, both consequences of the selection to reduce the variability of birth weight within-litter in the low line.

## Figures and Tables

**Figure 1 animals-10-00920-f001:**
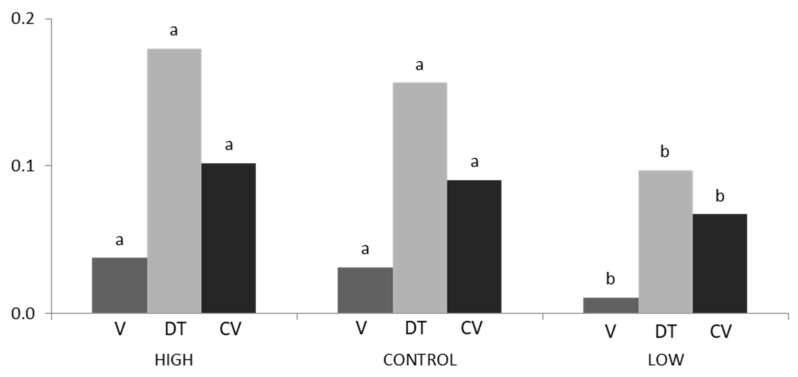
Variance (V), standard deviation (DT), and coefficient of variation (CV) means of birth weight within litter for high, low, and control lines. Different letters show statistically significant differences (*p* < 0.05).

**Figure 2 animals-10-00920-f002:**
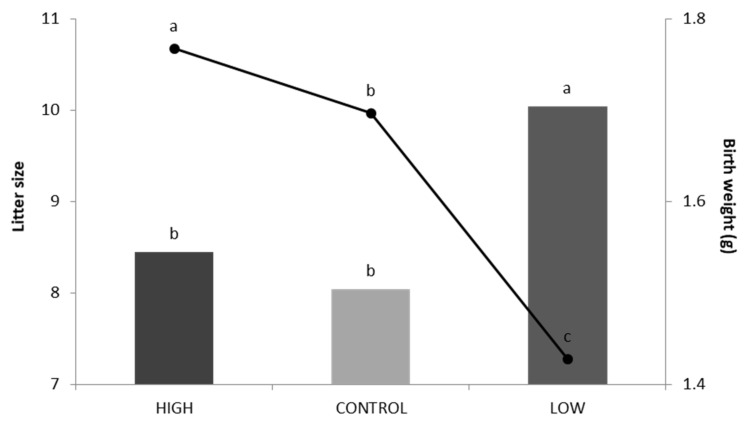
Litter size at birth (in bars) and birth weight (line) for the high line, control line, and low line. a,b,c show statistically significant differences (*p* < 0.05).

**Figure 3 animals-10-00920-f003:**
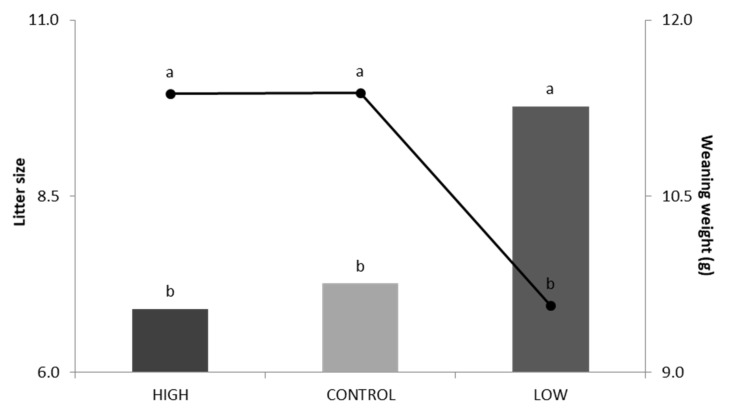
Litter size at weaning (in bars) and weaning weight (line) for the high line, control line, and low line. Different letters a,b,c show statistically significant differences (*p* < 0.05).

**Table 1 animals-10-00920-t001:** Number of records, dams, litters, and mean and standard deviation for the analysed traits in high variability line, low variability line, and control line.

	Trait	Records	Dams	Litters	Mean ± SD
Birth traits	BW	1697	132	192	1.58 ± 0.27
	LS	192	132	192	9.12 ± 2.81
	SB	192	132	192	95.38 ± 16.99
Weaning traits	WW	1602	132	192	10.79 ± 2.50
	LSw	192	132	192	8.26 ± 3.49
	WG	1602	132	192	9.20 ± 3.01
	SW	192	132	192	88.48 ± 26.66
Variability traits	V	186	129	186	0.025 ± 0.033
	SD	186	129	186	0.138 ± 0.075
	CV	186	129	186	0.085 ± 0.041

BW: birth weight (g); LS: litter size at birth (newborns); SB: survival at birth (%); WW: weaning weight (g); LSw: litter size at weaning (pups); WG: weight gain during lactation (g); SW: survival at weaning (%); V: variance (g); SD: standard deviation (g); CV: coefficient of variation.

**Table 2 animals-10-00920-t002:** Selection intensity (i), equivalent proportion selected (%), and effective population size in both high and low variability lines in all generations selected for birth weight environmental variability.

Generation	High			Low				
	i	%	Ne (F)	Ne (c)	i	%	Ne (F)	Ne (c)
Initial	1.93	7	111	85	−1.37	21	111	85
1	1.24	26	115	58	−1.19	29	119	63
2	1.16	30	119	51	−1.24	26	122	54
3	1.28	25	72	47	−1.33	23	110	51
4	1.23	27	68	46	−1.20	28	57	42
5	1.24	27	57	43	−1.08	34	50	39
6	1.46	18	54	39	−1.25	26	48	40
7	1.08	34	46	39	−0.90	44	50	40
8	1.32	23	46	39	−1.15	31	44	40
9	1.38	21	47	37	−1.23	27	46	38
10	1.15	31	41	36	−0.94	41	42	38
11	1.32	23	39	35	−1.13	31	40	36
12	0.80	50	41	34	−0.93	42	43	35
13	1.38	21	39	36	−1.01	38	41	37
14	0.94	41	39	35	−1.30	24	40	36
15	1.16	30	38	35	−1.23	27	40	36
16	1.22	28	40	35	−1.07	35	41	35
17	1.08	34	39	35	−1.34	22	40	35

Ne (F): effective population size computed from individual increase in inbreeding; Ne (c): effective size based on an increase in pairwise coancestry.

**Table 3 animals-10-00920-t003:** Significance of the difference between lines, parturition number, its interaction, sex, and linear and quadratic litter size covariate for the analysed traits.

	V	SD	CV	BW	WW	WG	LS	LSw	SB	SW
**Line**	***	***	***	***	***	***	***	***	*	*
**PN**	n.s.	n.s.	n.s.	***	***	***	*	n.s.	n.s.	n.s.
**Line*PN**	n.s.	n.s.	n.s.	***	n.s.	*	n.s.	n.s.	n.s.	n.s.
**Sex**	-	-	-	***	n.s.	n.s.	-	-	-	-
**LS**	n.s.	n.s.	n.s.	*	n.s.	n.s.	-	-	***	***
**LS*LS**	n.s.	n.s.	n.s.	n.s.	***	***	-	-	***	**

PN: parturition number; LS: litter size at birth; V: variance (g); SD: standard deviation (g); CV: coefficient of variation; BW: birth weight (g); WW: weaning weight (g); WG: weight gain during lactation (g); LSw: litter size at weaning; SB: survival at birth (%); SW: survival at weaning (%); ---: effect not taken into account; n.s.: not significant; * *p* < 0.05; ** *p* < 0.01; *** *p* < 0.001.

**Table 4 animals-10-00920-t004:** Mean and standard deviation (in brackets) for birth weight, weaning weight, and weight gain during lactation for line and parturition number.

Line	Parturition Number	BW	WW	WG
High	1	1.69 ^a^ (0.01)	11.78 ^a^ (0.11)	10.04 ^b^ (0.10)
	2	1.68 ^a^ (0.03)	10.98 ^b^ (0.21)	9.24 ^c^ (0.20)
Control	1	1.59 ^b^ (0.01)	12.10 ^a^ (0.09)	10.44 ^a^ (0.08)
	2	1.72 ^a^ (0.01)	10.67 ^bc^ (0.10)	8.91 ^c^ (0.16)
Low	1	1.38 ^c^ (0.01)	10.18 ^c^ (0.09)	8.74 ^c^ (0.08)
	2	1.39 ^c^ (0.02)	8.97 ^d^ (0.16)	7.52 ^d^ (0.10)

PN: parturition number; BW: birth weight (g); WW: weaning weight (g); WG: weight gain during lactation (g); ^a, b, c, d^: significant differences of the interaction between line and parturition number (*p* < 0.05).

**Table 5 animals-10-00920-t005:** Differences in survival at birth and at weaning and standard errors (in brackets) between lines.

Line	Survival at Birth (%)	Survival at Weaning (%)
High	88.68 ^b^ (±2.80)	79.64 ^b^ (±4.37)
Control	96.91 ^ab^ (±2.29)	87.91 ^ab^ (±3.57)
Low	98.47 ^a^ (±1.95)	96.37 ^a^ (±3.04)

^a, b^: significant differences between lines (*p* < 0.05).

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
