# Peer review of "Selection Response in a Divergent Selection Experiment for Birth Weight Variability in Mice Compared with a Control Line"

_animals, 2020, doi:10.3390/ani10060920_

Round 1

Reviewer 1 Report

Manuscript ID: animals-789620

Title: Selection response in a divergent selection experiment for birth weight variability in mice compared with a control line

General comments to manuscript

This manuscript presents an interesting and innovative topic such as the improvement of birth weight homogeneity. Recently, the homogeneity in weight at birth, at weaning and at slaughter within a batch has been considered as an important economic trait for prolific species like pigs and rabbits. This study can have important applications in selection programs of these species. The document is well written, and the experiment is well designed. I consider that this manuscript falls within the scope of this journal and recommend its publication with minor review.

Abstract

Pag     line

1          39        Please, insert “was observed in the low line than in the high line”       instead of “ was observed in the BW variability line than ………”  

Materials and Methods

The authors should give more details about the control population. For example, were the divergent lines obtained from the control population? i.e. Control, high and low lines have the same origin?

Has the control population been cryopreserved throughout these 17 generations of selection or did animals keep alive? If the control population was not cryopreserved, there could be genetic drift that would confuse the results, what do the authors think about it?.

Pag     Line

2          90                    Did the stillborns weigh? LS is the total number at births or live at birth?

3          92                    Variance and standard deviation give the same information, don’t they?

3          110                  A mixed model could be used to analyze more adequately BW, WW and WG, including dam as random effect.

Why wasn't survival at the birth analyzed?

SW is a ratio that includes the variable LS, could there be some spurious artifact when the LS is included in the model as covariate (LS + LS * LS)?

Discussion

Pag     Line

10        232-233          “…in the same direction…  “ Note that depends of traits, for example  the correlation between the mean weight and its variability shows a positive correlation, but for litter size is negative.

10        241                  Replace VBW for BW

10        251                  Replace the homogeneous line for the high line

10        257                  Please, replace the reference of a Congress [31] for one of a Journal  “Argente, MJ.; Calle, EW.; García, ML.; Blasco, A. Correlated response in litter size components in rabbits selected for litter size variability. J Anim Breed Genet2017, 134, 505– 511. https://doi.org/10.1111/jbg.12283”

10        258                  Replace     “five”              for                “seven”

11        275                    Insert “….in the weight at birth…..”

11        297                  Replace “pups” to “kits”

References

Change de references

[31]      To                    “Argente, MJ.; Calle, EW.; García, ML.; Blasco, A. Correlated response in litter size components in rabbits selected for litter size variability. J Anim Breed Genet. 2017; 134: 505– 511. https://doi.org/10.1111/jbg.12283”

[34]      To                   “ Argente, MJ.; García, M.L.; Zbyňovská, K.; Petruška, P.; Capcarová, M.; Blasco, A. Correlated Response to Selection for Litter Size Environmental Variability in Rabbits’ Resilience. Animal 2019; 13(10): 2348-2355. https://doi.org/10.1017/S1751731119000302

Author Response

Comments and Suggestions for Authors

Manuscript ID: animals-789620

Title: Selection response in a divergent selection experiment for birth weight variability in mice compared with a control line

Reviewer 1

General comments to manuscript

This manuscript presents an interesting and innovative topic such as the improvement of birth weight homogeneity. Recently, the homogeneity in weight at birth, at weaning and at slaughter within a batch has been considered as an important economic trait for prolific species like pigs and rabbits. This study can have important applications in selection programs of these species. The document is well written, and the experiment is well designed. I consider that this manuscript falls within the scope of this journal and recommend its publication with minor review.

ANSWER: We are grateful with the referee comment.

Abstract

Pag     line

1          39        Please, insert “was observed in the low line than in the high line”       instead of “ was observed in the BW variability line than ………” 

ANSWER: Done as suggested.

Materials and Methods

The authors should give more details about the control population. For example, were the divergent lines obtained from the control population? i.e. Control, high and low lines have the same origin?

ANSWER: Thank you for the comment. Yes, both control and selected lines have its origin in a common population. A clarification about this has been added in the data section (L83-85).

Has the control population been cryopreserved throughout these 17 generations of selection or did animals keep alive? If the control population was not cryopreserved, there could be genetic drift that would confuse the results, what do the authors think about it?.

ANSWER: The control line was not cryopreserved. We agree with the referee that genetic drift has affected this control line after elapsed generations. Fortunately this process is random in each line, and after 17 generations with a random genetic drift direction, it is reasonable to think that it has affected roughly the same in all the lines. This issue has been included in the revised version (L121-123).

Pag     Line

2          90                    Did the stillborns weigh? LS is the total number at births or live at birth?

ANSWER: Stillborns were no weighed but they were considered in the litter size. It has been clarified (L90-91).

3          92                    Variance and standard deviation give the same information, don’t they?

ANSWER: When the results are reported as a mean, mean of the variances is greatly affected by those litters with higher variance, providing a biased view. This is not the case when working on standard deviations, but, as the most usual indicator is the variance we have preferred to provide both parameters.

3          110                  A mixed model could be used to analyze more adequately BW, WW and WG, including dam as random effect.

ANSWER: We agree with the referee that a mixed model would be indicated in the case we were interested in individual genetic values, but the number of records in the control line is highly limited and would lead to errors. The genetic values for the control line would be regressed to the mean of the base population as there is no information of performances for this population linked to previous generation. In this case, as we are not interested in inferring about individual breeding values of animals, but only on the differences in the average performance, we consider that this model is more indicated in this scenario. This has been reinforced (L119-121)

Why wasn't survival at the birth analyzed?

ANSWER: Thank you for the suggestion. We have added analyses also for this trait. It has helped us to discover a bug in the computation of litter size at weaning. Survival at weaning has been affected and we have used to make other changes, such as including analysis of variance results for litter size at birth and at weaning and survival at birth. We are sure this will satisfy the curiosity of potential readers as for this reviewer. Modifications across the manuscript have been done accordingly.

SW is a ratio that includes the variable LS, could there be some spurious artifact when the LS is included in the model as covariate (LS + LS * LS)?

ANSWER: This is an interesting comment. Higher litter sizes increases both the numerator and the denominator of the survival and not proportionally, thus affecting the ratio but minimally in our opinion. In addition, it depends on the survival itself, as one death individual will be 0% of survival in a litter size of 1 and 90% in a litter of 10. However it will have the opposite effect the survival of one individual. From this it looks that the possible influence will of this dependence will affect more in the lower rank of the litter size, and more taking into account that also the quadratic covariate is fitted, reducing minimally the importance of this. However, the estimates of the covariates lead to consistent and reasonable conclusion that the optimal litter size was 10 from the survival point of view, a value in which this supposed influence would be negligible. We believe that this explanation is not worth expanding on. Instead, for the avoidance of doubt, we have added these optimal litter size values for survival (L212-213).

Discussion

Pag     Line

10        232-233          “…in the same direction…  “ Note that depends of traits, for example  the correlation between the mean weight and its variability shows a positive correlation, but for litter size is negative.

ANSWER: We apologized because this sentence is not clear enough. Here we are refereeing to the scale effect, the magnitude of the trait create an effect on the variability of the same trait in the same direction. The statistical scale effect can be defined as the relationship between the mean and the variability of a trait in the sense that the higher the mean of a variable, the higher its variability. We have modified the sentence accordingly (L241-242).

10        241                  Replace VBW for BW

ANSWER: There was a mistake in the sentence. It has been corrected.

10        251                  Replace the homogeneous line for the high line

ANSWER: We agree. Done as suggested.

10        257                  Please, replace the reference of a Congress [31] for one of a Journal  “Argente, MJ.; Calle, EW.; García, ML.; Blasco, A. Correlated response in litter size components in rabbits selected for litter size variability. J Anim Breed Genet. 2017, 134, 505– 511. https://doi.org/10.1111/jbg.12283”

ANSWER: Done as suggested.

10        258                  Replace     “five”              for                “seven”

ANSWER: Modified accordingly.

11        275                    Insert “….in the weight at birth…..”

ANSWER: Done as suggested.

11        297                  Replace “pups” to “kits”

ANSWER: We agree. Done as suggested.

References

Change de references

[31]      To                    “Argente, MJ.; Calle, EW.; García, ML.; Blasco, A. Correlated response in litter size components in rabbits selected for litter size variability. J Anim Breed Genet. 2017; 134: 505– 511. https://doi.org/10.1111/jbg.12283”

ANSWER: We thank the comment of the referee. The reference has corrected accordingly.

 [34]      To                   “ Argente, MJ.; García, M.L.; Zbyňovská, K.; Petruška, P.; Capcarová, M.; Blasco, A. Correlated Response to Selection for Litter Size Environmental Variability in Rabbits’ Resilience. Animal 2019; 13(10): 2348-2355. https://doi.org/10.1017/S1751731119000302

ANSWER: We thank the comment of the referee. The reference has corrected accordingly.

Reviewer 2 Report

Overall this is a well written paper that clearly explains it’s goals and results succinctly. There are a few sentences that confused me:

“Compared to the control line, a much higher response was observed in the BW variability line than in the high line...”

If this sentence is comparing the low BW variability line to the high BW variability line then it should read “observed in the low BW...” Other wise it’s meaning is unclear and needs to be clarified.

“This was extremely satisfactory given the homogeneity advantages in terms of animal welfare and robustness.”

Confusing sentence. Are you saying that the low variance lines are advantageous to animal welfare? If so it should read “given that homogeneity advantages” If not the sentence needs to be reworked to make it’s meaning clear.

“However, it was not proven if the result was balanced between the lines, or one of the lines did not react or the lines had responded with different magnitude.”

Confusing sentence. What are you saying here?

Author Response

Comments and Suggestions for Authors

Manuscript ID: animals-789620

Title: Selection response in a divergent selection experiment for birth weight variability in mice compared with a control line

Reviewer 2

Comments and Suggestions for Authors

Overall this is a well written paper that clearly explains it’s goals and results succinctly. There are a few sentences that confused me:

“Compared to the control line, a much higher response was observed in the BW variability line than in the high line...”

If this sentence is comparing the low BW variability line to the high BW variability line then it should read “observed in the low BW...” Other wise it’s meaning is unclear and needs to be clarified.

ANSWER: We apologized for the misunderstanding. We have clarified it now (L39).

“This was extremely satisfactory given the homogeneity advantages in terms of animal welfare and robustness.”

Confusing sentence. Are you saying that the low variance lines are advantageous to animal welfare? If so it should read “given that homogeneity advantages” If not the sentence needs to be reworked to make it’s meaning clear.

ANSWER: We agree. The text has been modified as suggested by the referee (L19-20, L40).

“However, it was not proven if the result was balanced between the lines, or one of the lines did not react or the lines had responded with different magnitude.”

Confusing sentence. What are you saying here?

ANSWER: We have clarified the sentence (L223-224).